# Sentinel Lymph Node Impact on the Quality of Life of Patients with Endometrial Cancer

**DOI:** 10.3390/jpm13050847

**Published:** 2023-05-17

**Authors:** Virginia García-Pineda, Alicia Hernández, Sonia Garrido-Mallach, Elena Rodríguez-González, María Alonso-Espías, Myriam Gracia, Rocío Arnedo, Ignacio Zapardiel

**Affiliations:** 1Gynaecologic Oncology Unit, La Paz University Hospital—IdiPAZ, 28046 Madrid, Spain; virginia.garciapineda@gmail.com (V.G.-P.); ignaciozapardiel@hotmail.com (I.Z.); 2Obstetrics and Gynaecology Department, La Paz University Hospital—IdiPAZ, 28046 Madrid, Spain; elenarogon@gmail.com (E.R.-G.);

**Keywords:** endometrial cancer, nodal assessment, sentinel lymph node biopsy, overall quality of life, sexual quality of life

## Abstract

Objective: Given the improvement in the surgical treatment of endometrial cancer with the inclusion of sentinel lymph node biopsy (SLNB), our aim was to evaluate the impact of this minimally invasive and tailored nodal assessment on patients’ quality of life (QoL). Methods: This was a cross-sectional study conducted in a single-centre, tertiary-level hospital. Patients diagnosed with preoperative early-stage endometrial cancer, who underwent primary surgical treatment between August 2015 and November 2021, were included. The enrolled patients were divided into two cohorts according to the nodal staging performed: the first group underwent only SLNB (SLNB group); the second group underwent pelvic and/or para-aortic lymphadenectomy (LND group). We evaluated the overall QoL using the European Organisation for Research and Treatment of Cancer (EORTC) Quality of Life core 30-item questionnaire (EORTC QLQ-C30) and a sexual health questionnaire (EORTC SHQ-C20). The scores were compared between the groups. Results: Ninety patients were enrolled in the study: 61 (67.8%) in the SLNB group and 29 (32.2%) in the LND group. In the LND group, 24 (82.7%) patients underwent pelvic and para-aortic LND, while 5 (17.3%) patients underwent pelvic LND. The assessment of the functional scales showed better results for the SLNB group than for the LND group, with a significantly lower impact on physical status (8.2% vs. 25%, respectively; *p* = 0.031). In terms of the symptom scales, the SLNB group reported a significantly lower negative impact on sleep quality (4.9% vs. 27.6%, respectively; *p* < 0.01), pain (1.6% vs. 13.8%, respectively; *p* = 0.019), and dyspnoea (0% vs. 10.3%, respectively; *p* = 0.011) than the LND group. The SLNB group had better results for all analysed items regarding sexual QoL. Conclusions: The implementation of a surgical technique with SLNB improved patients’ overall QoL by increasing their well-being in the functional and symptom spheres.

## 1. Introduction

Endometrial cancer is the most common gynaecological malignancy in developed countries, with an increasing incidence worldwide. In the United States, 65,950 cases were diagnosed in 2022, with excellent survival data. More than 65% of patients with endometrial cancer are diagnosed at stage I, with a 5-year survival rate of 95%, while the 5-year survival rate for all stages combined is 81% [1,2].

The standard surgical staging procedure for endometrial cancer has included extrafascial total hysterectomy with bilateral salpingo-oophorectomy and nodal assessment in most cases [1]. A minimally invasive surgical approach is associated with favourable oncological and surgical outcomes, particularly for specific patient groups, such as older women and women with obesity [3]. The therapeutic value of comprehensive lymphadenectomy of the pelvic and para-aortic areas has been controversial since the publication of two clinical trials, which demonstrated no impact on patient survival [4,5]. Given the morbidity associated with this technique and the absence of therapeutic benefit, sentinel lymph node biopsy (SLNB) has been postulated as the preferred method of lymph node assessment in early-stage endometrial cancer thanks to the published prospective evidence [6,7], as well as in cases of high-risk endometrial cancer [8,9].

SLNB is currently an acceptable option for nodal staging of endometrial cancer, according to the European guidelines and the National Comprehensive Cancer Network; however, there is a lack of information on the impact of SLNB on patients’ quality of life (QoL) [10,11].

Given that the treatment is curative for 80–90% of patients with stage I endometrial cancer, it is important not to lose focus on its impact on QoL. The main goal of gynaecological oncology in the past was to increase survival; however, the QoL of those years gained was not sufficiently considered.

QoL is defined by the World Health Organisation as “how individuals perceive their position in life, in the context of the culture and values in which they live, and in relation to their goals, expectations, aspirations and concerns”. Therefore, we need to consider these aspects of the patient beyond the disease-associated morbidity and mortality. The treatment of gynaecological cancer can produce deleterious effects, such as urinary and faecal incontinence, infertility, and altered body image, with a negative impact on the patient’s social life [12]. Moreover, surgical techniques and adjuvant radiotherapy can induce conditions such as lymphoedema and menopause, which can affect sexual desire, sexual intercourse, and overall QoL [13].

SLNB has been associated with a decreased risk of post-treatment lymphoedema compared with lymphadenectomy in patients who have undergone surgical staging for endometrial carcinoma. In the literature, the prevalence of lymphoedema ranges from 0% to 1.3% in SLNB and from 10% to 18% in lymphadenectomy [14]. In terms of major postoperative complications, the rate for lymphadenectomy was significantly higher than for SLNB (3.6% vs. 1.5%, *p* = 0.02), including a 2-fold increase in the risk of venous thromboembolism [14,15].

Current oncological management of endometrial cancer aims not only to reduce patients’ morbidity and mortality but also to provide them with the support of a healthcare team trained to care for all aspects of their lives affected by the disease and its treatment. Many survivors of gynaecological cancer live with discomfort, disfigurement, loss of activity, and reduced QoL due to lymphoedema. For certain patients, this morbidity is exacerbated by recurrent infections and hospitalisations, leading to deterioration and eventual disability [16].

Therefore, we consider it appropriate to explore the role of SLNB in the surgical treatment of endometrial cancer in our patients’ QoL. Our study’s main objective was to evaluate the QoL of patients with early-stage endometrial cancer, according to the nodal assessment method performed.

## 2. Methods

### 2.1. Sample and Design

This was a cross-sectional study conducted in a single-centre, tertiary-level hospital. Patients diagnosed with preoperative early-stage endometrial cancer who underwent primary surgical treatment between August 2015 and November 2021 were included in the study. Ninety patients who visited the Gynaecological Oncology Department for surveillance and agreed to participate in the study were included. Patients with any other synchronous malignancy or suspicion of recurrence were excluded. We obtained informed consent from all the patients, as well as ethics committee approval for the study (Ref. #PI-3676).

The included patients were grouped into two cohorts according to the type of nodal assessment performed during the primary surgical treatment. The first patient group underwent only SLNB (SLNB group), while the second group underwent pelvic and/or para-aortic lymphadenectomy with or without SLNB (LND group). We obtained the patients’ demographic and clinical information from interviews and their medical records, which included the tumour characteristics, the definitive International Federation of Gynaecology and Obstetrics (FIGO) stage of the disease, the type of surgery performed, the adjuvant therapy undergone, and the time elapsed since its completion.

### 2.2. Quality Assessment and Data Extraction

We assessed the overall QoL for all enrolled patients by employing the European Organisation for Research and Treatment of Cancer (EORTC) Quality of Life Core 30-item questionnaire (EORTC QLQ-C30). The 30 items in the EORTC QLQ-C30 covered functional scales (physical status and emotional, cognitive, and social roles), symptom scales (fatigue, nausea/vomiting, and pain), and other items (dyspnoea, insomnia, loss of appetite, diarrhoea, constipation, and financial concerns) commonly reported by patients with cancer [17].

Higher scores on the functional and symptom scales indicate a negative impact on patients’ QoL. To compare the impact on QoL between the groups, we analysed the scores from the QoL questionnaires reported by each patient. Patients answered each question on a scale from 1 (“not at all”) to 4 (“very much”). The highest scores (3–4, “very much”), representing the negative impact on patients, were combined to analyse the impact on QoL.

To evaluate possible sexual dysfunction in the treated women, the patients were provided with the EORTC-validated sexual health questionnaire, EORTC SHQ-C20 [18], whose purpose was to assess the most important subscales of the patient’s sexual sphere: loss of sexual desire and activity, vaginal dryness, ability to obtain pleasure, the degree of satisfaction, and perceived pain during sexual intercourse. Only questions from the EORTC SHQ-C20 questionnaire that assessed the subscales of interest were considered for the study. Higher scores on the sexual scales “loss of sexual desire”, “vaginal dryness”, and “pain” indicated a negative impact on the patient’s sexual QoL (SQoL). However, higher scores for “orgasm”, “sexual activity”, and “satisfaction” indicated a lower impact on SQoL. The highest scores (3–4, “very much”) were grouped to analyse the impact on SQoL.

Demographic and clinical characteristics of the patients are shown in Table 1. Table 2 and Table 3 show the results for the percentage of patients who responded with the highest score (“very much”) in each group.

### 2.3. Data Analysis

The categorical variables were reported as absolute numbers and percentages, whereas the continuous variables were reported as median and range. Student’s *t*-test and the Chi-squared test were employed to compare the study groups for the quantitative and qualitative variables, respectively. A multivariate analysis was performed using logistic regression. All variables were compared between the groups and the significance threshold was set at 0.05 for all analyses. Kaplan–Meier curves were employed to estimate the probability of an event occurring during the post-treatment follow-up period. We applied this method to those symptom variables that had a significant negative impact on the patients’ QoL. Statistical analyses were performed with SPSS software version 25.0 (IBM Corp., Armonk, NY, USA).

## 3. Results

The study enrolled a total of 90 patients whose clinical and demographic features are presented in Table 1. The SLNB group included 61 (67.8%) patients, all of whom underwent lymph node staging with SLNB only. The LND group included 29 (32.2%) patients, 24 (82.7%) of whom underwent pelvic and para-aortic LND; 24 (82.7%) also underwent SLNB. Most patients in the SLNB group had an early FIGO stage (96.7% stage I–II), with a significantly higher rate of FIGO stage III-IV in the LND group than the SLNB group (3.3% vs. 24%, respectively; *p* < 0.01). Therefore, a higher percentage of the LND group underwent adjuvant therapy than the SLNB group (21.3% vs. 82.8%, respectively; *p* < 0.01). However, there were no significant differences in the adjuvant radiotherapy employed, with more patients in the SLNB group apparently undergoing vaginal brachytherapy (69.2% vs. 59.1%; *p* = 0.799), with no significant differences between the groups.

### 3.1. Results of the QoL Core 30-Item Questionnaire (EORTC QLQ-C30) by Group

The results of the EORTC QLQ-C30 questionnaire in each group are shown in Table 2. The assessment of the functional scale showed better results for the SLNB group, with less impact on physical, emotional, social, and daily activities. However, the only statistically significant difference observed was in the reduced ability to perform strenuous activity in the LND group compared to the SLNB group (25% vs. 8.2%, respectively; *p* = 0.031) (Figure 1). Symptom scales showed similarly poorer results for the LND group. In the comparative analysis, the factors affecting patient morbidity, such as age (mean age of 59 (33–85) years for the SLNB group and 63 (41–81) years for the LND group; *p* = 0.489) and body mass index (BMI; mean BMI of 29 (18–41) for the SLNB Group and 28 (20–40) for the LND group; *p* = 0.142), were homogeneously distributed between the groups. The SLNB group had a better sleep quality and reported fewer insomnia events than the LND group during the follow-up period (4.9% vs. 27.6%, respectively; *p* < 0.01) (Figure 2). Post-treatment pain was significantly more pronounced in the LND group than in the SLNB group (13.8% vs. 1.6%, respectively; *p* = 0.019) (Figure 3). The number of patients reporting dyspnoea during the follow-up was significantly higher in the LND group than in the SLNB group (10.3% vs. 0%, respectively; *p* = 0.011) (Figure 4). Lastly, the LND group reported more financial difficulties than the SLNB group due to the impact of the treatment on their physical condition (7.1% vs. 0%, respectively; *p* = 0.036). There were no statistically significant differences in overall QoL in the last week (excellent rating of 88.5% for the SLNB group vs. 100% for the LND group; *p* = 0.057), after a median follow-up time from the treatment of 26 (0–73) months.

### 3.2. Results of the Sexual Health Questionnaire (EORTC SHQ-C20) by Group

In terms of the sexual sphere, the SLNB group appeared to show slightly better results; however, we observed no significant differences between the groups (Table 3). Nevertheless, the LND group showed significantly less vaginal dryness than the SLNB group (13.6% vs. 41.5%, respectively; *p* = 0.024) in the univariate analysis, which was not confirmed in the logistic regression analysis. Additionally, there were no significant differences in the rate of vaginal dryness between the patients who did not undergo vaginal brachytherapy and those who did (42.5% vs. 13%, respectively; *p* = 0.116).

## 4. Discussion

### 4.1. Summary of Main Results

Our study showed that the use of SLNB in the nodal assessment of patients with early-stage endometrial cancer had a lower impact on patients’ QoL than full lymphadenectomy. This comparative study between the 2 nodal assessment procedures showed that SLNB had significantly less impact on patients’ functional and symptom scores than lymphadenectomy. The patients who underwent SLNB reported a better performance status, i.e., they had significantly less difficulty performing strenuous activities than those in the LND group. In addition, the patients who underwent SLNB reported better sleep quality with less insomnia and significantly less post-treatment pain. Lastly, the impact on the patients’ physical condition and the treatment consequences produced significantly greater financial problems for the LND group than for the SLNB group. Despite the results, the impact on overall QoL was not significantly different between the groups after a median follow-up time from the treatment of 26 (0–73) months.

### 4.2. Results in the Context of Published Literature

Currently, there is a lack of information on the impact of SLNB inclusion on the nodal assessment of patients’ QoL. Given that surgery is the cornerstone of endometrial cancer treatment, we searched the literature for trials that evaluated the impact of various surgical aspects on QoL [19]. SLNB was associated with a reduced risk of post-treatment lymphoedema compared with complete lymphadenectomy in patients undergoing surgical staging for endometrial cancer [19,20]. Leitao et al. evaluated the prevalence of lower extremity lymphoedema (LEL) after SLNB versus complete lymphadenectomy and reported that the prevalence of LEL was 27% versus 41%, respectively (odds ratio 1.85, 95% CI 1.25–2.74).

Patients with self-reported LEL had significantly poorer QoL compared to those without self-reported LEL. Therefore, the authors concluded that SLN mapping provides accurate surgical staging, decreased morbidity, and improved QoL [21]. The GOG 244 study reported that symptoms following lymphadenectomy (e.g., heaviness, swelling, and numbness) had a negative impact on the women’s physical condition, especially during regular activities such as walking and performing strenuous activities. These adverse effects were associated with a greater impact on patients’ daily activities and could cause more episodes of pain and fatigue, which is consistent with our results in the LND group. These patients also experienced an increase in cancer-related distress and reported more symptoms of worry and emotional problems, which could be due to their reduced QoL being a constant reminder that the disease could return at any time during the follow-up period [22]. All of these results are consistent with our results in the LND group. In addition, this group reported poorer sleep quality and more insomnia during the follow-up, probably due to the increased distress from their disease. The literature also shows that women experienced poor body image and poorer sexual/vaginal health after lymphadenectomy [13]; however, no differences in the rates of sexual activity were noted for those with and without symptoms [23]. Similarly, our results showed a better SQoL for the SLNB Group, although the differences with the LND group were not significant.

The study by Tekbas et al., which compared symptoms in patients with gynaecological cancer before and after treatment, found significantly more severe symptoms of insomnia and dyspnoea in patients who had undergone treatment than in those who had not [23,24]. The symptom severity reported by the LND group in our study was consistent with that reported by Tekbas et al.

Other risk factors, such as obesity and age, could explain the observed differences between the groups on the functional and symptom QoL scales. The study by Karatasli et al. investigated the impact of BMI on the QoL of patients with endometrial cancer and concluded that the group with morbid obesity had poorer physical functioning than the group without morbid obesity (*p* < 0.011) [25]. However, we believe that the differences observed in our study were due to the surgical technique performed, given the homogeneity of the patient characteristics.

With regard to SQoL after treatment for endometrial cancer, several studies in the literature have demonstrated the occurrence of sexual dysfunction in treated patients [26,27], as we found in our LND group. A prospective controlled study published by Aerts et al. found differences in sexual function between patients with endometrial cancer and healthy controls after surgical treatment. Following surgery, cancer survivors reported lower sexual desire (*p* < 0.01), increased dyspareunia (*p* < 0.01), and decreased orgasmic intensity (*p* < 0.01) [28]. These results can be superimposed on those reported by the LND group. However, although the SLNB group reported better scores for these items, the difference was not significant.

### 4.3. Strengths and Weaknesses

Several publications comparing morbidity between the two methods of nodal assessment have been published in the past decade; however, there are no comparative studies on the impact of these techniques on the patients’ QoL [20,29]. Our study is a preliminary contribution to this research line, which will be explored in two ongoing randomised clinical trials [29,30]. The main weaknesses of our study are the lack of randomisation and the lack of multiple QoL assessments, mainly in the first months after treatment.

### 4.4. Implications for Practice and Future Research

The impact of SLNB on the QoL of our patients is still unknown, given the recent changes in the nodal assessment of early-stage endometrial cancer. Our study showed a lower impact on the functional and symptom scales than classical lymphadenectomy, meaning that this new technique improves the QoL of the treated patients. However, prospective studies with regular assessments of patients’ QoL over a longer follow-up period are needed to obtain more accurate results and assess the patients’ gradual adaptation to their new physical condition after treatment.

## 5. Conclusions

The recent inclusion of SLNB in the nodal assessment of patients with early-stage endometrial cancer showed a lower impact on patients’ QoL compared to patients who underwent complete lymphadenectomy, with SLNB demonstrating a significantly lower impact on the patients’ functional and symptom scales than complete lymphadenectomy.

## Figures and Tables

**Figure 1 jpm-13-00847-f001:**
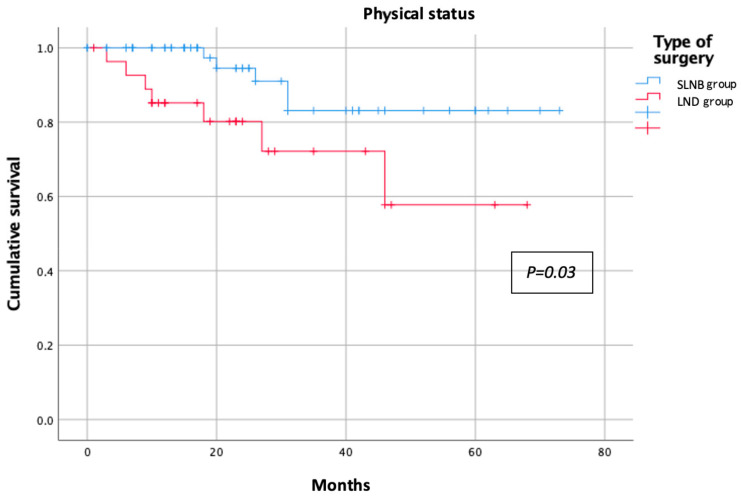
Physical status according to nodal staging.

**Figure 2 jpm-13-00847-f002:**
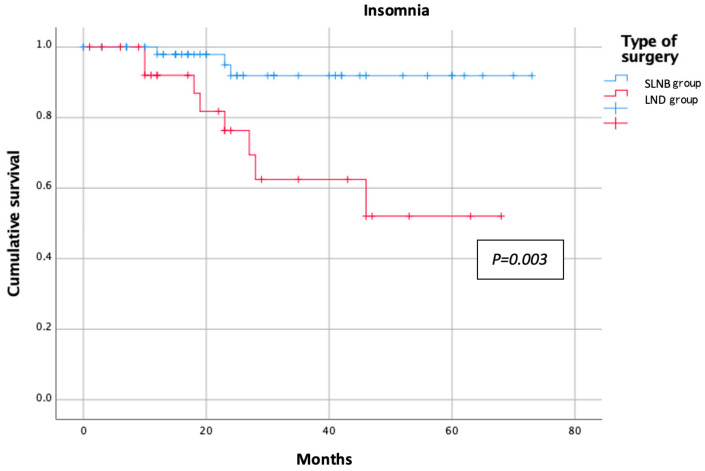
Sleep quality according to nodal staging.

**Figure 3 jpm-13-00847-f003:**
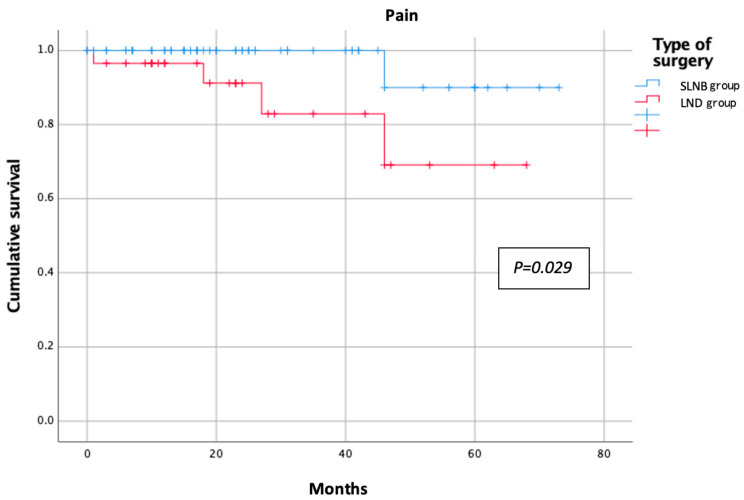
Reported pain according to nodal staging.

**Figure 4 jpm-13-00847-f004:**
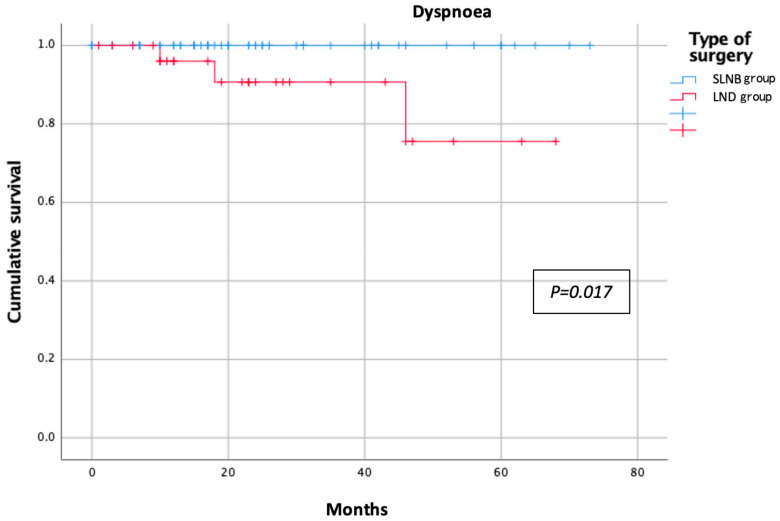
Symptomatic dyspnea according to nodal staging.

**Table 1 jpm-13-00847-t001:** Clinical and Demographic Features of the Study Population (*n* = 90).

Variables	SLNB Group *n* = 61 (%)	LND Group *n* = 29 (%)	*p*
Age in years, median (range)	59 (33–85)	63 (41–81)	0.489
BMI in kg/m^2^, median (range)	29 (18–41)	28 (20–40)	0.142
Type of surgery			
- TAH + BSO + SLNB - TAH + BSO + SLNB + Pelvic ± Para-aortic LND	61 (100) 0	0 29 (100)	**<0.001**
Surgical approach			
- Laparoscopy - Laparotomy	60 (98.4) 1 (1.6)	26 (89.7) 3 (10.3)	0.061
FIGO stage			
- I-II - III-IV	59 (96.7) 2 (3.3)	22 (76) 7 (24)	**0.002**
Grade			
- Low (G1–2) - High (G3)	60 (98.4) 1 (1.6)	19 (65.5) 10 (34.5)	**<0.001**
Administration of adjuvant treatment	13 (21.3)	24 (82.8)	**<0.001**
Radiotherapy			
- VBT - EBRT - EBRT + VBT	9 (69.2) 0 4 (30.8)	13 (59.1) 1 (4.5) 8 (36.4)	0.799
Chemotherapy administration	3 (4.9)	11 (37.9)	**<0.001**
Follow-up in months, median (range)	26 (0–73)	25 (1–68)	0.753

BMI, body mass index; TAH + BSO, total abdominal hysterectomy and bilateral salpingo-oophorectomy; SLNB, sentinel lymph node biopsy; LND, lymph node dissection; FIGO, International Federation of Gynaecology and Obstetrics; EBRT, external-beam radiotherapy; VBT, vaginal brachytherapy; kg, kilograms; m, meters.

**Table 2 jpm-13-00847-t002:** EORTC QLQ-C30 scores according to the type of nodal staging. The results are presented in absolute numbers and percentages.

*Eortc Qlq-C30*	Items	SLNB Group *n* = 61 (%)	LND Group *n* = 29 (%)	*p*
*Functional scales ^a^*	**Physical performance:**	** Very much **	** Very much **	
	1. Do you have any trouble doing strenuous activities?	5 (8.2)	7 (25)	**0.031**
	2. Do you have any trouble taking a long walk?	6 (9.8)	5 (17.2)	0.316
	3. Do you have any problem taking a short walk outside of the house?	1 (1.6)	1 (3.6)	0.568
	4. Do you need to stay in bed or a chair during the day?	1 (1.6)	2 (7.1)	0.182
	5. Do you need help with eating, dressing, washing yourself or using the toilet?	0 (0)	0 (0)	
	** Daily activities: **	** Very much **	** Very much **	
	6. Were you limited in doing either your work or other daily activities?	0 (0)	0 (0)	
	7. Were you limited in pursuing your hobbies or other leisure time activities?	0 (0)	1 (3.4)	0.145
	** Emotional functioning: **	** Very much **	** Very much **	
	21. Did you feel tense?	5 (8.3)	4 (13.8)	0.423
	22. Did you worry?	5 (8.3)	5 (17.2)	0.212
	23. Did you feel irritable?	3 (5)	1 (3.6)	0.764
	24. Did you feel depressed?	8 (13.8)	2 (6.9)	0.342
	** Cognitive functioning: **	** Very much **	** Very much **	
	20. Have you had difficulty in concentrating on things like reading a newspaper or watching television?	0 (0)	1 (3.6)	0.141
	25. Have you had difficulty remembering things?	3 (5.3)	1 (3.6)	0.729
	** Social functioning: **	** Very much **	** Very much **	
	26. Has your physical condition or medical treatment interfered with your family life?	1 (1.7)	1 (3.6)	0.577
	27. Has your physical condition or medical treatment interfered with your social activities?	1 (1.7)	2 (7.1)	0.187
*Symptom scales ^b^*	** Fatigue: **	** Very much **	** Very much **	
	10. Did you need to rest?	1 (1.6)	3 (10.3)	0.061
	12. Have you felt weak?	1 (1.7)	2 (6.9)	0.200
	18. Were you tired?	5 (8.3)	4 (13.8)	0.423
	** Nausea and vomiting: **	** Very much **	** Very much **	
	14. Have you felt nauseated?	0 (0)	1 (3.4)	0.148
	15. Have you vomited?	1 (1.6)	0 (0)	0.488
	** Pain: **	** Very much **	** Very much **	
	9. Have you had pain?	1 (1.6)	4 (13.8)	**0.019**
	19. Did your pain interfere with your daily activities?	4 (6.7)	3 (10.3)	0.546
	** Dyspnoea: **	** Very much **	** Very much **	
	8. Were you short of breath?	0 (0)	3 (10.3)	**0.011**
	** Insomnia: **	** Very much **	** Very much **	
	11. Have you had trouble sleeping?	3 (4.9)	8 (27.6)	**0.002**
	** Appetite loss: **	** Very much **	** Very much **	
	13. Have you lacked appetite?	1 (1.6)	0 (0)	0.488
	** Constipation: **	** Very much **	** Very much **	
	16. Have you been constipated?	4 (6.7)	4 (13.8)	0.271
	** Diarrhoea **	** Very much **	** Very much **	
	17. Have you had diarrhoea?	1 (1.7)	1 (3.4)	0.604
	** Financial difficulties **	** Very much **	** Very much **	
	28. Has your physical condition or medical treatment caused you financial difficulties?	0 (0)	2 (7.1)	**0.036**
*Overall health status ^c^*		** Normal-Excellent **	** Normal-Excellent **	
	29. How would you rate your overall health during the past week?	52 (85.2)	29 (100)	**0.029**
	30. How would you rate your overall quality of life during the past week?	54 (88.5)	29 (100)	0.057

Values in bold are statistically significant, *p* < 0.05. *^a^* High scores indicate high levels of adverse effects; *^b^* high scores indicate high levels of symptoms; *^c^* high scores indicate high satisfaction. The highest scores (3–4) are presented as “very much”. The number of patients who responded “very much” is expressed in absolute numbers and percentages. The main points in each sphere are underlined and in bold.

**Table 3 jpm-13-00847-t003:** *EORTC* Sexual Health Questionnaire according to nodal staging.

EORTC SHQ-C20 (Subscales)	SLNB Group *n = 61 (%)*	LND Group *n = 29 (%)*	*p*
	**Very much**	**Very much**	
Loss of sexual desire	22 (44)	14 (53.8)	0.415
Sexual activity	10 (17.5)	4 (13.8)	0.656
Vaginal dryness	17 (41.5)	3 (13.6)	**0.** **024**
Orgasm	20 (51.3)	9 (40.9)	0.436
Satisfaction	18 (47.4)	7 (33.3)	0.296
Pain	7 (16.7)	5 (21.7)	0.614

Values in bold are statistically significant, *p* < 0.05. The highest scores (3–4) are presented as “very much”. The number of patients who responded “very much” is expressed in absolute numbers and percentages.

## Data Availability

The QoL forms used in this study can be found at the following link: https://qol.eortc.org/questionnaires.

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
