# Peer review of "Sentinel Lymph Node Impact on the Quality of Life of Patients with Endometrial Cancer"

_jpm, 2023, doi:10.3390/jpm13050847_

Round 1
Reviewer 1 Report
This is a well written article. The background could be improved mentioning more studies that reported side effects of LND dissection versus SLN. Unfortunately, the number of enrolled patients is relatively small, but this is probably due to the single-center design of the study. Among the discussed symptoms of the QoL questionnaire, I would accentuate the relevance of PAIN, since chronic pain together with lyphoedema is one of the biggest complaints of patient with side effects from LND dissection, significantly impacts QoL and it is has been reported in literature. Very nice the Results in the context of Literature section.
just review some spelling
Author Response
Dear Reviewer 1,
First of all, I would like to thank you for considering this article for correction. In order to improve the background of the main topic of the article, I have added more references and expanded the information on the effect of lymphadenectomy on pain perception. I will be using a proofreading service for the minor English language editing that is required.
Your corrections are very much appreciated and I will submit the manuscript as soon as the final corrections have been made.
Kind regards,
Virginia
Reviewer 2 Report
Manuscript written by Virginia et al. is a valuable work, they aimed to evaluate the impact on patients’ quality of life (QoL) of this sentinel lymph node biopsy (SLNB) and tailored nodal assessment. However, several aspects should be taken into account. Please see below:
1. The author mentioned they used three questionnaires (global QoL by the European Organization for Research; Treatment of Cancer (EORTC) quality of life core 30-item questionnaire (EORTC QLQ-C30); sexual health questionnaire (EORTC SHQ-C20). Even though the author had already mentioned some detail of these questionnaires, I still suggest the author could put the whole questionnaires as supplements in this paper. Which could provide more information for the reader.
2. I suggest the author could summary all these 4-survival analysis figures into 1 figure and add a figure legend.
3. In the results part, I suggest the author could discuss these questionnaires and survival results into different section instead to put them into a whole section.
4. There are some language errors and typo. The authors should be revised the manuscript with an English language editor to make it more readable.
Extensive editing of English language required.
Author Response
Dear Reviewer 2,
First of all, I would like to thank you for your consideration and dedication in proofreading this article.
I will respond to your suggestions in the order in which they were mentioned:
1. The number of questionnaires we used is two: the European Organisation for Research and Treatment of Cancer (EORTC) Quality of Life Core 30-item questionnaire (EORTC QLQ-C30) and the EORTC validated sexual health questionnaire (EORTC SHQ-C20). In order to make more of these questionnaires available, their bibliographic references will be added so that the reader can access them. If there is no conflict of ownership, we will add them to the article as an appendix, following your recommendation.
2. Thank you very much for your suggestion, we will take it into consideration.
3. We will discuss the results of the impact on quality of life under several headings.
4. We will proofread the translation with an expert proofreader.
Thank you very much for your input, we will upload the latest version as soon as we have completed all the corrections.
Round 2
Reviewer 2 Report
The authors have addressed all my comments.